# Emotional Management Strategies in Prehospital Nurses: A Scoping Review

Marisa Almeida [1,*] , Catarina Lobão [2,*] , Adriana Coelho [2,3] and Vítor Parola [2,3]

1   Centro Hospitalar do Baixo Vouga, 3810-164 Aveiro, Portugal
2   Health Sciences Research Unit: Nursing (UICISA: E), Nursing School of Coimbra (ESEnfC),
    3000 Coimbra, Portugal; adriananevescoelho@esenfc.pt (A.C.); vitorparola@esenfc.pt (V.P.)
3   Portugal Centre for Evidence-Based Practice: A Joanna Briggs Institute Centre of Excellence (PCEBP),
    3000 Coimbra, Portugal
*   Correspondence: marisa.almeida.72080@chbv.min-saude.pt (M.A.); catarinalobao@esenfc.pt (C.L.)

**Abstract:** Background: Prehospital care is intricate and unpredictable. Nurses in this environment encounter psychologically challenging situations and traumatic experiences daily. Nurses respond variably when delivering care in this context. This study aims to map the emotional management strategies used by prehospital nurses. Methods: Research was conducted in MEDLINE, CINAHL Complete, and the Open Access Scientific Repository of Portugal (RCAAP). The inclusion criteria were studies with prehospital nurses in Portuguese, English, and Spanish languages, covering all study types. Results: From the initial 511 studies identified, four studies were deemed eligible after rigorous screening. The strategies used are individual (pre- and post-event) and collective, varying from formal to informal, with or without institutional support. Notable experiences included a lack of technical/scientific preparation, personal life association, treating acquaintances, pediatric-age patients, childbirth, cardiopulmonary arrests in young individuals, traffic accidents, and suicides. Conclusions: Nurses need training in basic emotional management tools. This research provides an initial understanding of their emotional well-being's impact on personal and professional performance. This study was prospectively registered with the Open Science Framework (OSF) on 29 June 2023, with the registration number: z638t.

**Keywords:** nurses; adaptation; psychological; emotional regulation; prehospital care; review

## 1. Introduction

Nursing care in a prehospital context represents a facet of nursing activity requiring diverse knowledge, abilities, and competencies. The prehospital emergency team is often the first differentiated aid to arrive in scenarios where circumstances can momentarily or permanently condition the lives of individuals and their families [1]. The care provided is highly challenging, as it unfolds in environments with unpredictable and adverse conditions, limited resources, and tension, which demands immediate and quality intervention, upon which the restoration of human life often depends [2]. In this vein, the prehospital emergency team, especially nurses, face daily psychologically draining situations and/or traumatic experiences, sometimes triggering strong emotions and/or stress reactions [3].

Specifically, the role of nurses in prehospital settings extends beyond just providing immediate medical intervention. They act as the initial triage, assessing the urgency and nature of care needed. They are also responsible for stabilizing patients, administering first aid, and ensuring patients' safe transport to healthcare facilities. This requires rapid decision-making, often with limited information, all while maintaining the emotional well-being of themselves, the patient, and accompanying relatives. The significance of emotional intelligence and management becomes paramount in this high-pressure, emotionally

charged environment. Their role's inherent unpredictability and emotional intensity highlight the crucial need to understand and prioritize their emotional well-being and the strategies they employ [4].

On the one hand, nurses might find their job rewarding and feel personally and professionally satisfied during care provision. On the other hand, the literature extensively documents feelings such as anxiety, distress, physical and emotional exhaustion resulting from negative experiences, or even feelings of powerlessness and frustration for not achieving expected outcomes, like recovering a person's health in critical situations [5]. Burnout, a syndrome arising from chronic workplace stress that has not been successfully managed, is a potential outcome of these chronic negative experiences. It is characterized by feelings of energy depletion, increased mental distance from one's job, or feelings of negativism or cynicism related to one's job, and reduced professional efficacy [6]. The relationship between emotional management, resilience, and burnout can be nuanced. While emotional management and resilience can act as protective factors, reducing the likelihood or severity of burnout, they are distinct from burnout itself, which is an outcome of prolonged stress and emotional exhaustion.

Some professionals seem to cope better than others: some can work in high-pressure situations for years; others leave the profession for less stressful environments [7]. Therefore, these professionals must be highly qualified and adopt suitable emotional management strategies to maintain optimal care quality and safety perceptions.

Prehospital care's unpredictability and fast pace often give nurses little time to recover from these negative experiences, traumatic events, or stress-inducing situations [8]. However, it is essential to discuss how they react and the impact on their emotional well-being and their personal and professional performance. Here, concepts of emotional intelligence and emotional management arise, with the latter being integral to the former.

While there might appear to be a confluence between emotional management and resilience, it is essential to restrict them for clarity. Emotional management pertains explicitly to identifying, understanding, and effectively modulating emotions in varied situations. Resilience, on the other hand, represents the broader ability to adapt to and recover from adversities. Burnout, distinct from both, is a potentially detrimental outcome when there is a failure in effective emotional management and resilience over extended periods. Considering that, at times, "resilience" and "emotional management" are used interchangeably in the literature, we integrated "resilience" into our search terms. Nonetheless, this review's main emphasis remains on the emotional management strategies used by prehospital nurses [9].

Emotional intelligence is a relatively recent research field aiming to broaden the traditional concept of intelligence, encompassing aspects related to the world of emotions and feelings. Daniel Goleman argues that it is impossible to separate rationality from emotions as they underpin decision-making efficiency, which is grounded in their control [10]. Thus, it is a crucial competency for nurses because establishing a therapeutic relationship requires identifying and understanding their emotions and those of the patient and their family. Awareness of these emotions is vital in optimizing personal and professional performance and care quality. Nurses must effectively manage the emotions that arise from being in continuous contact with vulnerability, illness, grief, and death to maintain their emotional stability and well-being [11,12].

The first references to emotional intelligence date back to 1920 with Edward Thorndike. Later, in 1990, Salovey and Mayer defined emotional intelligence as a set of skills related to perceiving, expressing, and regulating emotions in oneself and others. These skills allow us to control our emotions and assist others with controlling theirs, thus better guiding daily thoughts and actions [13].

In 1997, the same authors presented an emotional intelligence model explaining emotional information processing through a system of four interacting skills/components: emotion perception, emotional thought facilitation, emotional understanding, and emotional management [13]. Among these, emotional management, the study's most visible aspect, refers to the ability to reflexively regulate emotions in oneself and others, promoting

emotional and intellectual development. It involves skills such as receptiveness to pleasant and unpleasant feelings; the ability to intentionally engage or distance oneself from an emotion based on its utility; the ability to regulate one's and others' emotions clearly and assertively, without minimizing or accentuating; and the ability to reduce negative emotional states and increase positive ones [13]. Thus, Salovey and Mayer developed a comprehensive emotional intelligence theory. They suggest a concept of intelligence that processes emotions and benefits from them, implying the skill to think about emotions and reason with them, aiming to help individuals face the world and succeed. They explain emotional intelligence as the mental mechanisms needed for problem-solving and behavior management [13].

Daniel Goleman, in 2014, expanded on Salovey and Mayer's definition, defining emotional intelligence as the ability to recognize our feelings (self-awareness) and those of others (empathy), motivate ourselves (motivation), manage our emotions (self-management), and relate (social skills). Thus, emotional intelligence encompasses understanding one's emotions, empathizing with others, and controlling one's emotions to enhance life quality [13]. Based on this model, emotional intelligence enables individuals to make assertive daily choices by identifying sensations and recognizing, accepting, and managing emotions [13,14].

Emotional management is the foundation of personal and professional performance since emotions drive environmental adaptation, influencing cognitive processes such as perception, thinking, decision-making, language, beliefs, motivation, learning, memory, behaviors, attitudes, and intentions [15]. These significantly condition people's lives, so their daily journey is facilitated only if they acquire effective coping strategies.

Emotional skills influence the nurse's well-being, their relationships with colleagues, and the technical and human quality of care provided to end-of-life patients [16]. Caring for critically ill patients, those at the end of life, and dealing with death are inevitable aspects of nursing. Therefore, professionals require high levels of emotional intelligence to manage their continuous contact with death and loss [17]. Scientific evidence in this field is still minimal. It should constitute a research field due to the phenomenon's importance and its impact on nursing clinical practice and nurses' emotional well-being.

In a context where the nurse's role is predominant and nurse-person-family/caregiver interactions are constant, the emotional state of the nurse is constantly changing. Providing nursing care, especially in the prehospital environment, places nurses in front of problematic situations characterized by unpredictability, uncertainty, and disorder. Nurses who have clear feelings about their emotions and the situations they experience and can cope with them have lower stress levels at their workplace. Those who show a greater ability to reduce their negative emotional states and prolong the positive ones also experience higher overall health levels than those who struggle to manage their emotions [18].

Understanding and managing emotions is a fundamental skill in nursing and a professional requirement of humanized practice. Therefore, nurses with more developed emotional intelligence perform their tasks better because they can deal with emotions, mainly those aroused by these [19].

In recent years, various factors have led organizations to emphasize aspects related to emotional competencies. Once there may have been the consensus that emotions are the primary source of motivation because they trigger, sustain, and maintain human action [20]. In this regard, even Daniel Goleman in 2014 highlighted that health professionals' training should include some essential "tools" of emotional intelligence, especially self-awareness, and the arts of empathy and listening. Empathy is crucial in providing care centered on responding to others' needs. Training in emotional management (during training and throughout one's career) is paramount for nurses. If they do not possess or develop skills in this area, the consequences of negative experiences will cause damage, sometimes irreversible, altering their well-being and negatively interfering with their personal and professional performance [21].

*Aim*

The significance of effective emotional management in prehospital settings is well-established, as highlighted above. However, emotional management strategies are scattered in the scientific literature. After researching in MEDLINE (via PubMed), CINAHL Complete (via EBSCOhost), Joanna Briggs Institute (JBI) Database of Systematic Reviews, Cochrane Database of Systematic Review, PROSPERO, and Open Science Framework (OSF), it was found that there are currently no systematic reviews (published or underway) on this topic. Thus, while aiming to map the scientific evidence related to the emotional management strategies used by the prehospital nurse, a systematic literature review of the scoping review type was conducted addressing the following questions:

- What emotional management strategies are used by prehospital nurses?
- What are the characteristics of prehospital nurses' emotional management strategies?
- What experiences induce using emotional management strategies in nurses working in a prehospital context?

## 2. Materials and Methods

A scoping review allows for mapping the main concepts that support a knowledge area by summarizing and disseminating research data and identifying gaps in existing research, providing an overview of the existing evidence [22]. Thus, it is a methodology suitable for this research's aim and review questions, having been conducted by the guidelines of the JBI [22].

### 2.1. Inclusion Criteria

The eligibility criteria for the studies were defined based on the PCC mnemonic (participants, concept, context) in line with the methodology proposed by the JBI [22].

Concerning the participants (P), studies that considered individuals of both genders with the profession of the nurse were included. Regarding concept (C), studies on emotional management strategies were sought. As previously defined, emotional management strategies pertain to how individuals recognize, understand, and regulate their emotions or those of others in the face of psychologically taxing situations, traumatic experiences, or challenges. As for context (C), studies were included where care provision referred to emergency care outside the hospital setting.

Additionally, this review considers quantitative, qualitative, and mixed-method studies, as well as literature reviews, dissertations, and the grey literature. Regarding language, studies in Portuguese, Spanish, and English were considered.

### 2.2. Search Strategy

The search strategy utilized the electronic databases MEDLINE (via PubMed) and CINAHL Complete (via EBSCOhost). The search for unpublished studies included the Open Access Scientific Repository of Portugal (RCAAP).

The research was carried out in three phases. Initially, a search was conducted in the databases MEDLINE (via PubMed), CINAHL Complete (via EBSCOhost), and RCAAP, aiming to identify the most commonly used words in the titles and abstracts of the studies, as well as the indexing terms. The initial search keywords and phrases used were "nurs*", "emotional management", "emotional intelligence", and "prehospital". Subsequently, the identified words and terms were combined into a single search strategy and adjusted according to the specificities of each database/repository included in the review. Table 1 displays the search strategy, with the last access being on 14 March 2023. To conclude, each included work's reference list was analyzed to incorporate potential additional studies.

**Table 1.** Database search strategy and results.

| Database | Query | Record Retrieved |
|---|---|---|
| MEDLINE (via PubMed) | (((nurses[MeSH Terms]) OR (nurs*[Title/Abstract])) AND (("Emotional management" [Title/Abstract] OR "Emotional intelligence" [Title/Abstract] OR "Emotional regulation" [Title/Abstract] OR "Managing stress" [Title/Abstract] OR "Coping strategies" [Title/Abstract] OR "Emotional experiences" [Title/Abstract] OR "Resilience" [Title/Abstract] OR "Psychosocial adaptation" [Title/Abstract] OR "Psychological stress" [Title/Abstract]) OR ("Emotional Intelligence" [Mesh] OR "Emotional Regulation" [Mesh] OR "Resilience, Psychological" [Mesh] OR "Adaptation, Psychological" [Mesh]))) AND (("Prehospital" [Title/Abstract] OR "Out-of-hospital" [Title/Abstract] OR "Emergency Mobile Units" [Title/Abstract] OR "Emergency Responders" [Title/Abstract] OR "Emergency Ambulance Service" [Title/Abstract] OR "Ambulance" [Title/Abstract]) OR ((((Mobile Emergency Units[MeSH Terms]) OR (Emergency Treatment[MeSH Terms])) OR (Emergency Responders[MeSH Terms])) OR (Emergency Medical Services[MeSH Terms]))) Filters: English, Portuguese, Spanish, MEDLINE | 446 |
| CINAHL Complete (via EBSCOhost) | (TI nurs* OR AB nurs* OR MH (Nurses OR Emergency Nursing OR Emergency Nurse Practitioners)) AND (TI ("Emotional management" OR "Emotional intelligence" OR "Emotional regulation" OR "Managing stress" OR "Coping strategies" OR "Emotional experiences" OR "Resilience" OR "Psychosocial adaptation" OR "Psychological stress") OR AB ("Emotional management" OR "Emotional intelligence" OR "Emotional regulation" OR "Managing stress" OR "Coping strategies" OR "Emotional experiences" OR "Resilience" OR "Psychosocial adaptation" OR "Psychological stress") OR MH (Emotional Regulation OR Emotional Intelligence OR Stress Management OR Coping Strategies) AND (TI ("Prehospital" OR "Out-of-hospital" OR "Emergency Mobile Units" OR "Emergency Responders" OR "Emergency Ambulance Service" OR "Ambulance") OR AB ("Prehospital" OR "Out-of-hospital" OR "Emergency Mobile Units" OR "Emergency Responders" OR "Emergency Ambulance Service" OR "Ambulance") OR MH (Prehospital Care OR Emergency Care OR Ambulances OR Emergency Medical Services)) Filters: English, Portuguese, Spanish, excluding MEDLINE | 50 |
| RCAAP | (Enfermeiro (assunto) AND gestão emocional (assunto)) OR (Enfermeiro (assunto) AND inteligência emocional (assunto)) OR (Enfermeiro (assunto) AND coping (assunto)) OR (Enfermeiros (assunto) AND pré-hospitalar (assunto)) | 15 |

## 3. Results

After the search was completed, the screening process began to identify the studies that met the eligibility criteria.

### 3.1. Study Selection

The results obtained through this process are presented in Figure 1 following the recommendations of "The PRISMA 2020 statement: an updated guideline for reporting systematic reviews" [23]. As depicted, the search identified 511 potentially relevant studies. Of these, 18 were excluded because they were duplicates; of the remaining 493 studies, 477 were excluded after the title and abstract assessment; and 12 out of the remaining 16 articles were excluded for not meeting the inclusion criteria after full-text reading. Ultimately, four studies were included in this review.

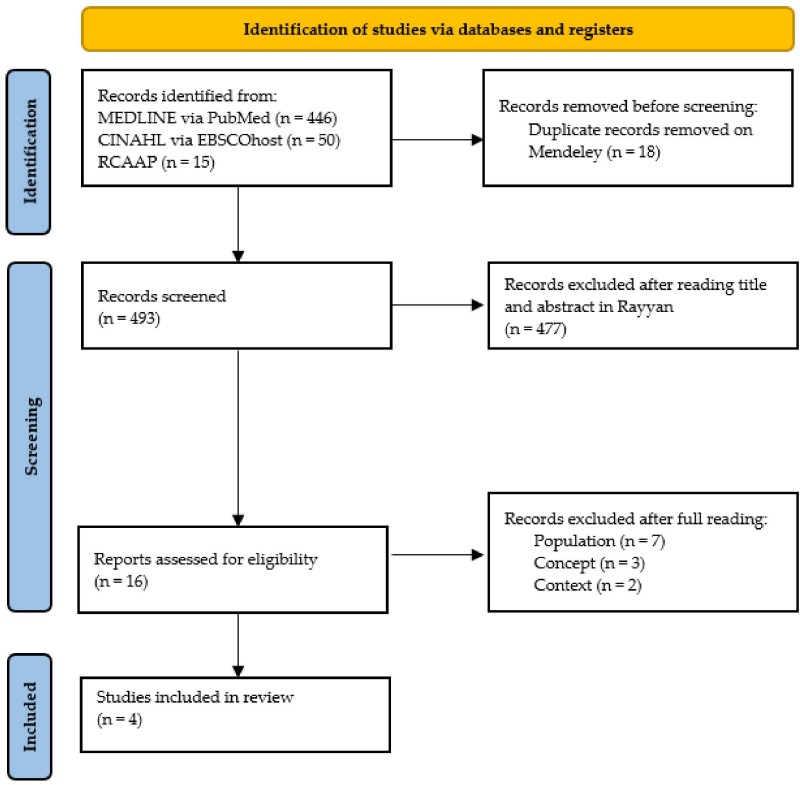

**Figure 1.** PRISMA flowchart of the study selection process [23].

### 3.2. Data Extraction

Data were extracted from the articles included in this scoping review by two reviewers independently. Any doubts and disagreements that arose were discussed, and a third reviewer was consulted. A pilot test was conducted by two independent reviewers. Initially, they analyzed the title/abstract, and subsequently, the full text. For the initial analysis, 5% of the total research was considered to achieve at least a 75% consensus between the reviewers. In the second phase of the study, 2% of the full-text articles were analyzed to maintain the same level of agreement. Studies that were part of this review were identified and included based on the established inclusion criteria and research limiters [22]. The protocol of this scoping review is registered on the OSF platform (https://osf.io/z638t —accessed on 29 June 2023) and can be consulted for more information.

The four studies, which are presented in Table 2, were designated as S1 [24], S2 [25], S3 [26], and S4 [27] for ease of identification.

**Table 2.** Summary of articles' findings.

| Study | Author Year Country | Title | Type of Study | Aims |
|---|---|---|---|---|
| S1 | Svensson and Fridlund (2008) Sweden [24] | "Experiences of and actions towards worries among ambulance nurses in their professional life: A critical incident study". | Qualitative with the use of semi-structured interviews | Describe the critical incidents that ambulance nurses experience throughout their professional lives and their actions to prevent and cope with them. |
| S2 | Bohström, Carlström, and Sjöström (2017) Sweden [25] | "Managing stress in prehospital care: Strategies used by ambulance nurses". | Qualitative with the use of semi-structured interviews | Identify the strategies used by ambulance nurses to cope with traumatic events. |

**Table 2.** *Cont.*

| Study | Author Year Country | Title | Type of Study | Aims |
|---|---|---|---|---|
| S3 | Carvello, Zanotti, Rubbi, Bacchetti, Artioli, and Bonacaro (2019) Italy [26] | "Peer-support: a coping strategy for nurses working at the Emergency Ambulance Service". | Qualitative with the use of semi-structured interviews | Explore prehospital nurses' experiences, opinions, and feelings regarding the use of the peer-support model. |
| S4 | Vicente, Jansson, Wickström, Danehorn, and Wahlin (2021) Sweden [27] | "Prehospital Emergency Nurses' coping strategies associated to traumatic experiences". | Qualitative with the use of semi-structured interviews | Identify the coping strategies used by prehospital nurses after exposure to traumatic events. |

The included studies were published between 2008 and 2021 regarding the temporal distribution. Geographically, the distribution is focused on Europe, with three of the studies conducted in Sweden. The studies adopt qualitative and predominantly descriptive approaches using semi-structured interviews, as this methodology is most suitable for exploring experiences, life events, and emotions.

### 3.3. Answering Review Questions

It is now essential to present the results of the studies included in Table 3 to answer the previously described review questions.

**Table 3.** Responses to study's review questions.

| Study | Sample | Emotional Management Strategies | Characteristics of Emotional Management Strategies | Experiences that Led to the Use of Emotional Management Strategies |
|---|---|---|---|---|
| S1 | 25 nurses | Individual | Ignore the concern | - Perception of lack of experience and preparation for more specific occurrences<br>- Specific emergencies that are associated with one's personal life<br>- Activation for pediatric-age victims<br>- Concerns about the work environment where safety, equipment, or trust in peers is compromised |
| | | | Reflection on the action by anticipating it or maintaining focus | |
| | | | Emotional management influenced by length of service/experiences | |
| | | | Seeking training and practice/simulation | |
| | | Collective | Request for support after the incident: dialogue with a trusted individual and debriefing sessions | |
| | | | Request for additional support for the emergency situation | |
| S2 | 15 nurses | Institutional support | Informal conversations | - Insufficiency: of information, resources, preparation, and control (births and pediatric-age victims, family and friends)<br>- Uncertainty: In life-threatening situations for the professional, significant previous experiences, lack of support, namely debriefing sessions |
| | | | Support from peers or experts | |
| | | | Informal debriefing at the end of the incident/shift | |
| | | Sharing of experiences | Confidences shared with colleagues | |
| | | | Advantages of teamwork | |

**Table 3.** *Cont.*

| Study | Sample | Emotional Management Strategies | Characteristics of Emotional Management Strategies | Experiences that Led to the Use of Emotional Management Strategies |
|---|---|---|---|---|
| S3 | 14 nurses | Self-care and coping | Emotional distancing: withdrawal, denial, minimization, focus on families and loved ones | - Aid to pediatric polytrauma victims, death due to cardiorespiratory arrest in young people, and management of communication with parents~ <br> - Traffic accidents and suicides |
| | | | Emotional decompression: crying, irony, sports | |
| | | Peer support | Dialogue with work colleagues with the possible inclusion of a support peer—'peer-support'—or a psychologist | |
| S4 | 12 nurses | Mental preparation increases a sense of control | Acceptance of the inevitability of rescue situations | - No data |
| | | | Recognition of moments of weakness and the need for psychological survival | |
| | | Knowledge and professionalism as a measure of control | A constant search for knowledge updating | |
| | | | Time and previous experiences are aspects that seem to intervene in emotional management favourably | |
| | | | Adoption of a professional role, avoiding the transmission of emotions and focusing energy on aiding the individual | |
| | | | Development of interprofessional work, both in task delegation and decision-making | |
| | | Healing/self-healing process | Release of emotions: Experiences need to be understood and processed, and thus, talking with work colleagues is a means | |
| | | | Dialogue/debriefing in the sense of processing the experience | |
| | | | Engagement in pleasurable activities such as spending time with family, walking, exercising, elevating mood | |
| | | | Acceptance of the situation by validating that they did everything possible and finding answers/explanations for the event | |
| | | | A positive view of their work and life, valuing what they have and gaining self-confidence | |

S1 is a qualitative study published in Sweden that identified emotional management strategies they termed as individual and collective. The individual strategies focused on

oneself, and the collective ones encompassed colleagues and the institution. It became clear that the perception of a lack of experience and technical preparation, concern for safety and equipment, association of the occurrence with personal life, or activation for pediatric-age victims are experiences that lead to emotional management strategies.

In S2, also qualitative and published in Sweden, the strategies used by prehospital nurses were identified, distinguishing between institutional support and sharing experiences. Institutional support in a formal aspect and the sharing of experiences in an informal way highlighted the benefits of teamwork. Situations where they experienced uncertainty and a lack of information, resources, preparation, and control, especially for childbirth and pediatric-age victims, relatives, and friends, and situations of life-threatening risk for the professional and previous significant experiences, were those where they most often used emotional management strategies.

S3, conducted in Italy and of a qualitative type, identified strategies such as self-care, coping, and peer support. The latter was further explored, aiming to include a support pair to promote ways of handling, understanding, and regulating emotions. Assisting poly-traumatized pediatric patients, encountering death from cardiopulmonary arrest in young individuals, managing communication with parents, and dealing with traffic accidents and suicides were perceived as the most demanding situations and were prone to mobilizing the pointed emotional management strategies.

Lastly, S4, with a qualitative approach and once again conducted in Sweden, introduced strategies such as mental preparation, knowledge, and professionalism as essential measures in the perception of control over events and the healing/self-healing process. The latter refers to ways of understanding and regulating emotions, either by releasing emotions through dialogue or by engaging in activities they find enjoyable. This study does not provide data that answer the questions about experiences that led to using the same strategies.

When reflecting on the findings from the studies, certain themes and patterns emerge that draw a cohesive picture of the emotional management strategies employed by prehospital nurses, as well as the unique nuances of each research work.

Considering the commonalities in the evidence mapped, all four studies (S1–S4) identify a blend of individual and collective strategies for emotional management. It is evident that there is a significant emphasis on personal responsibility and agency in managing emotions, whether these strategies are termed as self-care, mental preparation, or individual strategies. Concurrently, the value of a collective approach in managing emotions shines through, irrespective of whether this is framed as peer support, teamwork, or institutional support. Furthermore, numerous studies pinpoint specific demanding situations as catalysts for the employment of emotional management strategies. Particularly striking is the recurring reference to dealing with pediatric-age victims, highlighted in studies S1, S2, and S3. Additionally, traffic accidents and situations posing life-threatening risks to professionals emerge as common triggers, as reported in S2 and S3. A noteworthy observation is that three out of the four studies (S1, S2, and S4) were qualitative in design and carried out in Sweden. This could suggest potential cultural or regional consistencies in the challenges prehospital nurses face and the strategies they deploy in this country.

By delving into the distinctions among the studies, each offers unique insights. S1's central theme revolves around the dichotomy between individual and collective strategies. In contrast, S3 provides a more in-depth exploration of concepts like coping and peer support. S4 introduces the importance of mental preparation, knowledge, and professionalism in controlling events and fostering a healing/self-healing process. Varied perspectives also emerge regarding the challenges that prehospital nurses face. While S1 emphasizes perceived inadequacies, such as a lack of experience and technical preparedness, S3 underlines the complexities of managing communication with parents or confronting suicides. Another divergent theme is the perception of institutional versus informal support. In S2, a clear distinction is drawn between the formalities of institutional support and the more casual, yet profoundly beneficial, act of sharing experiences.

Through this synthesis, we gain a more holistic understanding of the strategies pre-hospital nurses adopt in managing their emotions, appreciating both the common threads that bind the studies and the unique tapestries each research paints.

## 4. Discussion

The global landscape varies regarding health professionals providing prehospital care, with two methodological paradigms: the European and American systems. The European system mainly relies on clinical professionals (doctors and nurses), preferably from areas in direct contact with critically ill patients (anesthesia, intensive care, and emergency). In contrast, the American system primarily uses emergency technicians and paramedics [28]. While our primary focus centered on the emotional management of prehospital nurses irrespective of the geographic context, it is notable that the studies we mapped emerged from Europe. This difference between the two systems is essential to provide context to the mapped studies in this scoping.

This European emphasis might be explained by the historical context in Portugal where the first mentions of prehospital nursing date back to the 1980s, with a progressive differentiation and indispensability. Currently, they represent a fundamental resource in the medicalized prehospital sector. This might explain why all the included studies were conducted in Europe, paralleling the target population (nurses). Thus, as presented, three of these studies were conducted in Sweden, highlighting the importance that the Swedish researchers place on the subject and emphasizing the role of nurses as specialized professionals in the European prehospital setting.

The prominence of European studies in this review does not indicate a regional bias but represents the available literature. It is essential to understand that the intention was not to juxtapose the European and American systems, but to shed light on the varying emotional management strategies nurses employ in prehospital care.

Furthermore, the significance of this topic also appears to be rising. Although the time distribution spans from 2008 to 2021, three studies are from post-2017. This indicates a growing concern for the emotional well-being of nurses.

All the studies adopt qualitative and predominantly descriptive approaches, using semi-structured interviews. The samples encompass groups ranging from 12 to 25 pre-hospital nurses, characterized by age, gender, educational level, professional experience, and experience in prehospital settings. Despite the advantages of qualitative research, especially given the researcher's direct interaction with individuals who experienced the studied phenomenon [29], it is essential to underline the absence of quantitative studies in this field, which could validate the obtained results.

Regarding the emotional management strategies used before an event, S1 describes that nurses often either disregard their concern about the situation or, on the contrary, reflect on the circumstances they might encounter, anticipating them and maintaining focus. S3 aligns with this, suggesting that nurses seek emotional distance by distancing themselves from the situation. S4 emphasizes adopting a professional role, avoiding the transmission of emotions, and focusing their energy on aiding the person in need. All these aspects highlight the necessity for mental preparation by nurses as an essential measure in their perception of control over incidents.

Most of the strategies can be grouped into individual and collective categories. As observed from those mentioned above, the emotional management strategies employed before incidents are individual. To these, one can add the length of service/life experiences as positively influencing emotional management, as mentioned in S1 and S4. S4 adds the acceptance of the inevitability of rescue situations, the acknowledgement of moments of weakness, the need for psychological survival, and a positive outlook on work and life—valuing what they have and gaining self-confidence—as facilitators of emotional management. S3 also touches upon emotional decompression through crying or, on the flip side, using irony or humor.

Regarding collective strategies, nurses seek conversations with colleagues or someone they trust in all studies and debriefing sessions after the event/shift in S1, S2, and S4 to share experiences. S2 and S3 emphasize that dialogue support can be sought from colleagues or specialists, with S3 highlighting the importance of including peer support. This concept refers to a colleague who becomes a support partner, having a more formal component (previously defined), or from a psychologist and/or doctor, with institutional support being crucial here. However, in S2, nurses report that only colleagues genuinely understand the circumstances of the cases, being more capable of making professional comments and providing feedback than specialists. Colleagues allow them to review situations, reflecting on what could have been done differently and offering mutual learning. S4 sees the conversation with colleagues as a means of releasing emotions since these need to be understood and processed, as mentioned.

This analysis also allows an understanding of emotional management strategies as formal and informal. Debriefing sessions can be held formally, as a regular institutional support measure, and informally, as a process of self-care and sharing experiences. All studies point to the informality of debriefing sessions and conversations between colleagues as a quick means and facilitating sharing experiences. Nurses invoke their importance regarding frequency, which should be more significant, formality/informality, and timing, preferably at the end of the event/shift, to manage their emotions. These sessions are crucial in processing the experience; they do not have the desired effect if they occur days after it. According to S2, nurses prefer to do it with the team that starts the shift and when the colleagues involved in the specific event are present to assess and process their experiences, analyzing the case from another angle. Trust in colleagues is also essential for the well-being of nurses: shared viewpoints and future challenges are seen as part of daily work, and the feelings and experiences shared under traumatic experiences create a unique connection, according to S2.

The lack of flexibility of institutions in adopting dynamics and managing the recovery time from events perceived as traumatic by nurses is also evident (S2). The request for extra/differentiated support for emergencies as a way to obtain more security in assistance (S1), the satisfaction and recognition of the advantages of teamwork, and the development of interprofessional work, both in task delegation and decision-making, in both S2 and S4 characterize the formal emotional management strategies.

The search for training and simulation/training as a way of constantly updating knowledge appears in S1 and S4, in response to the need they express to feel capable, with technical and scientific skills that allow them to respond to the variability and complexity of the events they find in the prehospital setting. In this way, knowledge and professionalism arise as a measure of preparation and a feeling of control. Formally, institutions should train, prepare, and train their professionals so that they can perceive safety and quality in their care.

The strategies used after the occurrence, except for informal conversations and formal debriefing sessions, are preferably individual. In S3, the nurses mention denying and minimizing the impact of the event they face on their emotional well-being. In turn, in S4, the nurses mention accepting the situation by validating that they did everything possible, finding answers/explanations for the event, and engaging in pleasurable activities such as spending time with family, walking, and exercising. In S3 and S4, the nurses mention using humor for self-care and healing/self-healing.

Alongside the characterization of emotional management strategies, various experiences led to their use and were reported by the nurses. S1 and S2 mention the perception of a lack of experience, technical preparation, and professional skills to manage and control more specific/complex occurrences. One study found that, in addition to the unpredictability of the prehospital environment, there is a lack of information and preparation for complex situations that increase the feeling of a lack of control, as pointed out here [30]. S2 highlights the insufficiency of information, lack of resources to identify the clinical diagnosis, or on the contrary, too much information to manage (like driving the ambulance,

waiting for the specialized team, or simply the difficulty in finding the address). S1 specifies emergencies associated with personal life and S2 mentions providing assistance to family members, friends, or colleagues.

S1, S2, and S3 agree when they suggest the difficulty in providing care to pediatric-aged victims during childbirth and deaths due to cardiopulmonary arrest in young people because of the complications and complexity of the procedures to be established. Another investigation supports this finding, indicating that care for the victims of cardiopulmonary arrest, especially children and young individuals, determines certain peculiarities, both in the immediate way of acting and in the subsequent impact on nurses' moods/emotions when the situation is not reversed [31]. S2 highlights the demanding task of managing communication, the therapeutic relationship with the family/caregiver in an emergency, and the need to demonstrate empathy simultaneously. In this context, S3 underscores the difficulty in managing communication with parents when faced with pediatric-aged victims, and adds traffic accidents and suicides as complex experiences to manage. The literature mentions delivering bad news and psychological support to the family as emotionally challenging moments [31]. Other authors have observed that caring for people in cardiopulmonary arrest generates stress, anxiety, and insecurity [32]. The impact of these occurrences on the well-being of nurses aligns with the findings of S1, S2, and S3.

S1 and S2 also raise concerns about the work environment where safety, equipment (technical failures), or trust in peers is questioned. Similarly, it is documented that communication failures [33] and deficient teamwork [32] elicit negative feelings in nurses. S1 and S2 also highlight situations of life-threatening risk for the professional, where their safety is jeopardized as they face exposure to alcohol, weapons, drugs, and threats of violence, or from previous significant experiences. The literature mentions the physical and emotional exhaustion linked to the ongoing concern with the work environment, especially regarding safety and the possibility of technical failures in a setting with limited resources [30]. The lack of support from management, as mentioned in S2, which does not grant time off after a traumatic experience, also leads to nurses resorting to various emotional management strategies.

It is crucial to also mention the limitations of this review. Restricting the research to three languages (Portuguese, English, and Spanish) because the researchers master them may have limited the inclusion of other pertinent studies. Concerning the population, since prehospital care involves a multi-professional team, the focus was exclusively on nurses, given that it is the researchers' field of expertise. Although this criterion meant excluding various studies, since many do not differentiate participants in the results section, it prevented the risk of biasing this study's result. While our review aimed to provide a comprehensive understanding of the emotional management strategies employed by nurses, a notable limitation is the inclusion of only four articles from the vast number of initial search results. This resulted from our stringent adherence to the predefined inclusion criteria to ensure high relevance and quality. Future reviews could consider more flexible inclusion criteria or a broader scope to encompass a more extensive set of studies.

## 5. Conclusions

Our review highlights findings from various studies, suggesting that nurses, both intentionally and, at times, unintentionally, employ emotional management strategies to navigate and process the emotions arising from their daily care-providing experiences. Emotions, as described in the literature, are perceived in multifaceted ways, making it imperative for nurses to acquire skills that help decode the effects of these emotions on their well-being and performance, whether they be positive or negative.

From the studies we examined, informal conversations and debriefing sessions are frequently mentioned as valuable tools for emotional management. This suggests a potential benefit for healthcare institutions to consider these methods in their daily professional dynamics. The mapped literature also indicates that nurses often experience stress, anxiety,

and perceived insecurity, especially in challenging situations like critical care or in the multifaceted prehospital environment.

Based on our review, there seems to be a consensus regarding the importance of equipping nurses with not just technical competencies, but also non-technical competencies. Along with their technical expertise, the literature points towards the potential advantages of training nurses in fundamental emotional intelligence and management skills, such as self-awareness, empathy, and active listening. The role of health unit management in promoting strategies that allow nurses to process and integrate their emotions is also emphasized in many of the studies we reviewed.

Additionally, our review identified a recurrent theme highlighting the benefits of technical and scientific preparation, especially when dealing with more complex care scenarios like pediatric care. Teamwork, trust, and the ability to manage emotions in challenging situations are frequently cited as key contributors to fostering self-confidence among nursing professionals.

All these, as suggested by the literature we reviewed, have the potential to prevent challenging events from escalating and adversely impacting nurses' well-being and performance across various timeframes. Our review shows that emphasizing the importance of understanding and implementing emotional management strategies, especially in prehospital settings, is essential. The literature consistently advocates for a deeper exploration of the psychological toll of demanding scenarios and emphasizes the value of research grounded in nurses' perceptions and the strategies they find effective. This focus can offer greater insight into their emotional responses, the challenges they face, and the impact of their emotional well-being on critical care delivery to individuals, families, or caregivers.

In summarizing the findings from the literature, it is evident that mapping prehospital nurses' emotional management strategies can offer insights that might inform future interventions and training. As the studies suggest, recognizing and effectively managing emotions remains a core tenet of nursing—a profession deeply rooted in compassionate and humane practice.

**Author Contributions:** Conceptualization, M.A., C.L. and A.C.; methodology, M.A., C.L. and A.C.; validation: M.A., C.L., A.C. and V.P.; investigation, M.A., C.L. and A.C.; resources, M.A., C.L., A.C. and V.P.; writing— original draft preparation, M.A.; writing—review and editing, M.A., C.L., A.C, and V.P.; visualization, M.A., C.L., A.C. and V.P.; supervision, C.L. and A.C. All authors have read and agreed to the published version of the manuscript.

**Funding:** This research received no external funding.

**Institutional Review Board Statement:** Not applicable.

**Informed Consent Statement:** Not applicable.

**Data Availability Statement:** Not applicable.

**Public Involvement Statement:** No public involvement in any aspect of this research.

**Guidelines and Standards Statement:** This manuscript was drafted against the "The PRISMA2020 statement: An updated guideline for reporting systematic reviews" for systematic reviews research.

**Conflicts of Interest:** The authors declare no conflict of interest.

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
