# Peer review of "Emotional Management Strategies in Prehospital Nurses: A Scoping Review"

_nursrep, doi:10.3390/nursrep13040128_

Round 1

Reviewer 1 Report

Comments and Suggestions for Authors

Thank you for the opportunity to review and provide feedback on this work. Below I have provided feedback for consideration organized by manuscript section for ease of review.

Abstract: Inclusion of your search results (e.g. 511 studies, etc) would be helpful to the reader to understand the magnitude of potentially available evidence on the selected topic

Introduction: A nice overview of the concepts of emotional management and emotional intelligence is provided. However, this section lacks basic information on the role of the nurse within prehospital care provision. Providing this detail aids the reviewer in having a better understanding of the role of nurses within this setting thus understanding how the concepts of emotional intelligence/management are related. Essentially incorporating this information and restructuring the introduction to make it clear to the reviewer why a review such as the one conducted is necessary would be helpful.

Further it seems emotional management is similar to resilience strategies and it would also be helpful for the reader to understand the differences (if differences exist) if not then resilience should be included as part of the search terms and the review revised to include this information.

Aim-Line 128 is a strong statement and I would not agree that the introduction provides "irrefutable" evidence that such a review is needed. Consider revising this statement with more neutral language. Further, a scoping review is intended not to answer specific review questions as stated in lines 137-140 but rather to map the available evidence in a field, identify key concepts/knowledge gaps, etc. Methodologically to answer the stated questions a systematic review such as an integrative review would have been more appropriate. An integrative review aims to synthesize existing empirical and theoretical knowledge relevant to the clearly defined problem.

Data extraction: It is not adequate to refer readers to a registered protocol a summary of the data extraction process should be included within the manuscript.

Results: This section reads like a summary of each article instead of a synthesis across all included articles. Consider revising to discuss the commonalities across study results, and highlight differences in study results. 

Discussion: Within this section there is a brief description of the differences between European and American prehospital care. This would be more beneficial within the introduction. Additionally, because the American system is so different does it make sense to blend both of these systems into a single review?

Overall this manuscript has potential to be a good contribution to the literature however there are some methodological issues that should be addressed. Further, it is not clear how the concepts described within this manuscript differ from those of resilience and burnout. This should also be addressed in subsequent revisions. Thank you for the opportunity to review and provide feedback. Best of luck in your revisions.

Comments on the Quality of English Language

There are several sentences throughout that are awkwardly phrased which should be revised for clarity prior to potential publication. A few examples include lines 30-32, lines 43-46, and 59-61 to name a few.

Author Response

Reviewer 1:

Dear Reviewer 1,

Thank you for taking the time to review our manuscript and for your valuable feedback. We appreciate your suggestion

Reviewer 1: Abstract: Inclusion of your search results (e.g. 511 studies, etc) would be helpful to the reader to understand the magnitude of potentially available evidence on the selected topic.

Authors response: We have now updated the "Results" section of the abstract to state, "From the initial 511 studies identified, four studies were found eligible after rigorous screening." This modification offers a concise yet comprehensive overview of our search results, aiding the reader's understanding. Line 18-19.

Reviewer 1: Introduction: A nice overview of the concepts of emotional management and emotional intelligence is provided. However, this section lacks basic information on the role of the nurse within prehospital care provision. Providing this detail aids the reviewer in having a better understanding of the role of nurses within this setting thus understanding how the concepts of emotional intelligence/management are related. Essentially incorporating this information and restructuring the introduction to make it clear to the reviewer why a review such as the one conducted is necessary would be helpful.

 – Authors response: We appreciate your observation about the lack of detail on the nurse's role within prehospital care provision. We agree that this context is crucial to fully understand the relevance of emotional intelligence/management in this setting. To address this, we have incorporated a paragraph that delves into the specific roles and challenges nurses face in prehospital care. This added section connects their unique position to the necessity of emotional intelligence and management, thereby providing a stronger foundation for the subsequent discussion in the manuscript. Line 39-48.

Reviewer 1: Further it seems emotional management is similar to resilience strategies and it would also be helpful for the reader to understand the differences (if differences exist) if not then resilience should be included as part of the search terms and the review revised to include this information.

Authors response: Thank you for highlighting the importance of distinguishing between emotional management and resilience. Upon revisiting our search strategy, we can confirm that "Resilience" was indeed incorporated into our search terms. We included resilience as a search term, recognizing that, in some contexts, it is used interchangeably with emotional management.

Our primary aim, however, is to delve deep into the realm of emotional management strategies utilized by prehospital nurses. To address your concern, we have enriched the introduction to elucidate the nuances between emotional management and resilience. We've emphasized that emotional management focuses on the immediate recognition, understanding, and modulation of emotions, while resilience encompasses a broader spectrum of adaptability in the face of adversity. We trust this provides clarity on the subject and ensures our readers remain informed about our review's primary focus. Line 71-80

Reviewer 1: Aim-Line 128 is a strong statement and I would not agree that the introduction provides "irrefutable" evidence that such a review is needed. Consider revising this statement with more neutral language. Further, a scoping review is intended not to answer specific review questions as stated in lines 137-140 but rather to map the available evidence in a field, identify key concepts/knowledge gaps, etc. Methodologically to answer the stated questions a systematic review such as an integrative review would have been more appropriate. An integrative review aims to synthesize existing empirical and theoretical knowledge relevant to the clearly defined problem.

Authors response: We appreciate your insights regarding the nature of our review and the language employed in articulating its aim.

1 - In response to your comment about using the term "irrefutable", we acknowledge that it might come off as a strong assertion. Thus, We have modified the text to better reflect the substantial, yet not conclusive, evidence on the importance of emotional management in prehospital settings. Line 157-159.

2 - Concerning the methodology, we would like to clarify that while our review questions are specific, they align well with the intent of a scoping review. Our questions offer an explorative overview of the existing evidence, identifying key concepts, knowledge gaps, and characterizing the field's breadth and depth. Scoping reviews, by their nature, can have specific yet broad-encompassing questions aiming to map and understand the available research landscape. The questions we posed fit this intention, as they seek to provide a panoramic view of emotional management strategies in the prehospital nursing context.

We understand the distinction you highlighted between integrative and scoping reviews. An integrative review aims to synthesise empirical and theoretical knowledge around a clearly defined issue. On the other hand, our review strives to map the existing literature, explore primary concepts, identify patterns, and discern knowledge gaps in emotional management strategies for prehospital nurses. Hence, we believe a scoping review is the appropriate methodology for our research goals.

Reviewer 1: Data extraction: It is not adequate to refer readers to a registered protocol a summary of the data extraction process should be included within the manuscript.

Authors response: Thank you for highlighting the importance of transparency in the data extraction process.

Our research utilized a data extraction instrument outlined in the protocol. The instrument developed aligned well with our data extraction needs for this scoping review. However, we understand your point that referring readers to an external protocol may not be sufficient for comprehensive understanding within the context of the manuscript.

To address your concerns and enhance the manuscript's clarity, we are willing to include a concise summary of the data extraction process within the paper. This addition will provide readers with immediate insight into our approach without requiring them to reference the external protocol. Still, we believe that mentioning the protocol offers a layer of transparency and provides those interested with a more detailed methodological understanding. Line 222-236

Reviewer 1: Results: This section reads like a summary of each article instead of a synthesis across all included articles. Consider revising to discuss the commonalities across study results, and highlight differences in study results. 

Authors response: We understand the importance of synthesizing the findings to draw out commonalities and highlight differences among the included studies rather than presenting them as mere summaries.

In response to your comment, we have revised the Results section to integrate a discussion on the overarching themes and patterns that emerge from the studies alongside the unique nuances of each research work. We've endeavoured to balance the individual summaries of the studies (S1-S4) with a holistic reflection on shared and distinct themes. Through this revised synthesis, we hope to provide readers with a more comprehensive understanding of prehospital nurses' strategies in managing their emotions, appreciating both the shared themes and the unique insights each study brings. We believe this approach better serves the objective of our review and hope it meets your expectations. Line 282-314

Reviewer 1: Discussion: Within this section there is a brief description of the differences between European and American prehospital care. This would be more beneficial within the introduction. Additionally, because the American system is so different does it make sense to blend both of these systems into a single review?

Authors response: Our aim was to review emotional management strategies among prehospital nurses comprehensively. The preponderance of European studies is not a reflection of intentional focus but rather an outcome of the studies available in the literature. We recognize the potential differences and uniqueness of the American system, and we did not intend to draw direct comparisons between the two. However, the distinction between the two systems provides context and demonstrates the variations in prehospital care globally. Line 321-327; 334-337.

Reviewer 1: Overall this manuscript has potential to be a good contribution to the literature however there are some methodological issues that should be addressed. Further, it is not clear how the concepts described within this manuscript differ from those of resilience and burnout. This should also be addressed in subsequent revisions. Thank you for the opportunity to review and provide feedback. Best of luck in your revisions.

Authors response: Thank you. In response to your observation about the differentiation between the concepts described within the manuscript and those of resilience and burnout:

Differentiation from Resilience: As detailed in our introduction, we have delineated the distinctions between "emotional management" and "resilience." (as a previous comment you mention about this two concepts). We acknowledged the occasional interchangeability of these terms in the literature and ensured that our manuscript focuses primarily on emotional management strategies.

Inclusion of Burnout: Following your suggestion, we have enriched our introduction by integrating a discussion on "burnout.” Burnout, arising from chronic workplace stress, is a potential detrimental outcome when there's a failure in effective emotional management and resilience over extended periods. We have elaborated on how emotional management and resilience can act as protective factors against burnout, emphasizing their distinct roles and importance in prehospital nursing.

We believe these revisions provide a more precise understanding and differentiation of emotional management, resilience, and burnout concepts. We have also made several other improvements throughout the manuscript to enhance its quality and contribution to the literature.

Once again, we sincerely thank you for your insightful comments and guidance. We are optimistic that these revisions will make our manuscript a stronger contribution to the field. Line 54-61; Line 75-80.

Reviewer 2 Report

Comments and Suggestions for Authors

The article addresses an interesting issue: the emotional management of nurses' early response to patients. However, it gives the impression that this is an article written without the time needed to search and read the literature rigorously.

In general, the article claims an originality that is perhaps not so: a simple search in Scopus yields hundreds of results on other literature reviews focusing on the emotional dimension of nurses. In fact, there is also no citation of articles published in Nursing Reports and close to the chosen topic. Nor is there any mention of covid-19 and its impact on the emotional stability of healthcare workers.

In this sense, the conclusions begin by stating that "The present investigation demonstrated that nurses intentionally and often unintentionally employ emotional management strategies (...)". It is paradoxical to read such strong statements in the case of a review study, i.e., based on what other empirical studies have said. Such statements should be qualified.

In any case, the main problem observed is the paucity of articles used as a sample. It is hard to understand that from a search that yields hundreds of results, only four articles are selected. Taking only four studies as sources of information to answer such ambitious research questions is quite striking. This suggests that perhaps the inclusion criteria used should have been made more flexible.

Finally, the discussion with the results obtained in other studies is rather scarce (it is concentrated in only two paragraphs).

This leads me to recommend to the authors that they should thoroughly revise the article before resubmitting it.

Comments on the Quality of English Language

Authors should review English grammar in general, in order to refine certain expressions.

Author Response

Reviewer 2:

Reviewer 2: The article addresses an interesting issue: the emotional management of nurses' early response to patients. However, it gives the impression that this is an article written without the time needed to search and read the literature rigorously.

In general, the article claims an originality that is perhaps not so: a simple search in Scopus yields hundreds of results on other literature reviews focusing on the emotional dimension of nurses. In fact, there is also no citation of articles published in Nursing Reports and close to the chosen topic. Nor is there any mention of covid-19 and its impact on the emotional stability of healthcare workers.

Authors response: Thank you for your thoughtful feedback and the time invested in reviewing our manuscript. Your insights have been valuable, and we aim to comprehensively address each of your concerns.

Rigor of Literature Search: We recognize and appreciate your observation about the perceived rigor of our literature search. Our scoping review methodology was designed to capture a broad spectrum of literature, and indeed, our initial searches identified hundreds of results. However, as you rightly pointed out, identifying potential articles is one thing while ensuring that they meet specific inclusion criteria is another. As is typical with scoping reviews, our methodology aimed to provide an overview of the existing literature on the subject.

Originality and Contextualization: Regarding the apparent originality of our review, we understand your perspective. While there may be a multitude of studies that examine the emotional dimensions of nurses, our focus has been on emotional management strategies within prehospital settings, a niche that hasn't been widely explored. Nonetheless, in light of your feedback, we have added clarifications in our manuscript to differentiate our study from other reviews and to emphasize its unique contribution. Line 71-80

Inclusion of Relevant Publications: We deeply value the ethos of "Nursing Reports" and appreciate its impartial stance on including its own publications in submitted works. Our literature selection was based on content relevance, strictly adhering to the scope of our scoping review, rather than the specific journals in which they were published. This approach prioritizes the content and quality of the studies over the name or prestige of the publishing journal. However, we did include a reference from "Nursing Reports" in our revised manuscript, showcasing our commitment to encompassing the most pertinent and relevant literature available. Reference 6.

Reviewer 2: In this sense, the conclusions begin by stating that "The present investigation demonstrated that nurses intentionally and often unintentionally employ emotional management strategies (...)". It is paradoxical to read such strong statements in the case of a review study, i.e., based on what other empirical studies have said. Such statements should be qualified.

Authors response: We recognize and appreciate your concerns regarding the assertiveness of our initial statements. In response to your observation, we have revised the discussed section to better align our conclusions with the nature of our review study, emphasizing that our findings are based on existing literature rather than primary empirical evidence. We have modified our language to indicate that the strategies and insights presented are suggested by the literature we reviewed.Line 461-531.

Reviewer 2: In any case, the main problem observed is the paucity of articles used as a sample. It is hard to understand that from a search that yields hundreds of results, only four articles are selected. Taking only four studies as sources of information to answer such ambitious research questions is quite striking. This suggests that perhaps the inclusion criteria used should have been made more flexible.

Authors response: Thank you for your thoughtful feedback on the number of articles included in our review. We understand and acknowledge your concerns regarding the perceived limited sample of articles, especially considering the initial large number of search results.

While our search yielded hundreds of results, the final selection of only four articles directly resulted from our strict adherence to the predefined inclusion criteria. Our intent was to ensure that the selected articles closely matched the specific research questions and objectives of our scoping review. We recognize that this limited number may raise concerns about comprehensiveness, but our primary focus was to ensure the relevance and quality of the studies included. We have also acknowledged this aspect in the discussion section as a limitation to convey the scope and boundaries of our review transparently. Line 454-459.

Reviewer 2: Finally, the discussion with the results obtained in other studies is rather scarce (it is concentrated in only two paragraphs).

Authors response: We recognize the value of a comprehensive discussion to provide context and richer interpretation of our findings. In response to this feedback, both the results and discussion sections have been revised and expanded to offer a more detailed account of our findings and their alignment with the broader body of literature. We believe these enhancements will provide a clearer perspective on what has been mapped out in our scoping review. Line 282-314; Line 321-326; Line 334-337.

Round 2

Reviewer 2 Report

Comments and Suggestions for Authors

From my point of view, the manuscript has been sufficiently improved to warrant publication in Nursing Reports.